# Analysis of Changes in Land Use/Land Cover and Hydrological Processes Caused by Earthquakes in the Atsuma River Basin in Japan

Yuechao Chen *  and Makoto Nakatsugawa

Water Environment System Laboratory, Muroran Institute of Technology,
Mizumoto 27-1, Muroran 0508585, Japan; mnakatsu@mmm.muroran-it.ac.jp
* Correspondence: 19096505@mmm.muroran-it.ac.jp; Tel.: +81-070-4020-6754

**Abstract:** The 2018 Hokkaido Eastern Iburi earthquake and its landslides threaten the safety and stability of the Atsuma River basin. This study investigates land use and land cover (LULC) change by analyzing the 2015 and 2020 LULC maps of the basin, and its impact on runoff and sediment transport in the basin by using the soil and water assessment tool (SWAT) model to accurately simulate the runoff and sediment transport process. This study finds that the earthquake and landslide transformed nearly 10% of the forest into bare land in the basin. The simulation results showed that the runoff, which was simulated based on the 2020 LULC data, was slightly higher than that based on the 2015 LULC data, and the sediment transport after the earthquake is significantly higher than before. The rate of sediment transportation after the earthquake, adjusted according to the runoff, was about 3.42 times more than before. This shows that as the forest land decreased, the bare land increased. Conversely, the runoff increased slightly, whereas the sediment transport rate increased significantly in the Atsuma River basin after the earthquake. In future, active governance activities performed by humans can reduce the amount of sediment transport in the basin.

**Keywords:** 2018 Hokkaido Eastern Iburi earthquake; land use and land cover (LULC) change; SWAT model; runoff; sediment transport; Atsuma River basin

## 1. Introduction

Over 11,400 massive earthquakes with an intensity of Mw 6 or higher have occurred globally between 1 January 1900, and 31 December 2020 [1,2]. As the global population rapidly increases, the threats posed by earthquakes and secondary disasters on human lives, property, and infrastructure are expected to increase [3]. Over 10,000 earthquakes occur each year in Japan, given that the Japanese archipelago is located at the junction of the Pacific plate and the Asia–Europe plate, where earthquakes occur frequently. Landslides caused by severe earthquakes often destroy large amounts of landscape vegetation, resulting in significant changes in runoff potential and sediment transport [4]. Considerable geomorphic instability is often present, along with extensive erosion, deposition, and bank line shifting of the river basin after a strong earthquake. Furthermore, earthquakes cause massive configuration changes to the landmass, while the river's course shifts [5]. Such problems can severely affect safety and stability, as well as the process of runoff and sediment transport in river basins.

LULC change, primarily attributed to human activities [6] and natural disasters [7], is recognized as one of the most important components of global environmental change [8,9]. It is also one of the most important drivers of hydrological processes, as it influences all available water resources and flow regimes in river basins worldwide [10]. LULC change not only affects the river flow, but also the other components of the hydrological process, such as sediment transport [11]. Therefore, assessing the impact of LULC change on hydrology is essential for watershed management and ecological restoration. Consequently,

the quantification of the impact of LULC change on the dynamics of streamflow in river basins has been an area of interest for hydrologists in recent years. However, while most studies have focused on the influence of LULC change caused by rapid urbanization, deforestation, and agriculture, only a few discuss the impact of earthquakes and other natural disasters. Unlike human activities, natural disasters, such as earthquakes, have a significantly larger impact on LULC change over a short period of time.

There are some studies on the 2018 Hokkaido Eastern Iburi earthquake. Susukida et al. [12] investigated the tectonic stress field in and around the aftershock area of the earthquake and found that the reverse fault-type stress field was dominant in the aftershock area. Zhou et al. [13] analyzed the mechanism of regional landslides and the stability and permanent displacement of slopes based on the effects of continuous heavy rainfall and seismic motion. Shibata et al. [14] reported changes in groundwater levels based on the responses of the groundwater level to the M2 tidal constituent before and after the earthquake. Kubo et al. [15] investigated the source rupture process of an earthquake in Japan and reproduced the overall ground motion characteristics of the sedimentary layers of the Ishikari Lowland. Li et al. [16] investigated the controlling role of the Ta-d pumice, which significantly influences the coseismic landslides that are triggered by an earthquake. Ohtani et al. [17] assessed the seismic potential around earthquakes that would occur in the future. Results showed increasing seismic risk. Gou et al. [18] investigated the seismic structures and complex seismic azimuthal anisotropy in the source area of an earthquake. Fujiwara et al. [19] identified and analyzed the surface displacements associated with earthquakes. Nakamura et al. [20] investigated the S-wave attenuation (Qs) structure in and around Hokkaido, Japan, including a consideration of the source area of the earthquake and its aftershocks. Fukuda et al. [21] analyzed the sleep pattern data of junior high school students on the night of a blackout after an earthquake, comparing their sleep to a normal night. Chen et al. [22] made a rough estimate of the sediment transport before and after the earthquake, and qualitatively judged that the sediment transport increased greatly after the earthquake. Although research on earthquakes has been in-depth and comprehensive, only a few researchers have quantified LULC change after an earthquake and studied its impact on the hydrological processes of the Atsuma River basin. In this study, we analyze LULC changes caused by an earthquake and its impact on hydrological processes.

To further improve our understanding of the impact of land use change on the runoff and sediment transport processes in the Atsuma River basin, we adopted the SWAT model for our investigation. Previously, the SWAT model has been used to calculate rainfall runoff and snowmelt runoff [23,24], and in research on sediment [25], nutrients [26], pollution and microbial transport processes [27], as well as regional water resource management [28] and other research fields. Furthermore, the SWAT model has been used to evaluate the impact of LULC change on runoff and sediment transport processes. Babur et al. [29] investigated the effects of climate and LULC change on sediment yield at the Mangla Dam. Perazzoli et al. [30] analyzed the effects of changes in stream flow and sediment yield under different LULC scenarios in the Concordia River basin. Sadeghi et al. [31] investigated the runoff response to climate variables and LULC change in the Tajan River basin and found that LULC significantly impacted runoff compared to climatic variables. Anand et al. [32] assessed the hydrological regimes of the Ganga River basin through LULC change. These studies achieved good results, indicating that the SWAT model is suitable for evaluating the impact of LULC change on runoff and sediment transport processes in river basins. Robust techniques other than SWAT models are also applied to investigate the relationships between climate change and streamflow. For example, Ghaderpour et al. [33] applied least-squares cross-wavelet analysis to show the impact of climate change on snowmelt and streamflow in the Athabasca River basin in Canada. Zerouali et al. [34] also applied cross-wavelet transform analysis to assess the response of daily rainfall and karst spring discharge for the Sebaou River basin in northern Algeria. The specific objectives of this study are: (i) to study the rapid and drastic change of LULC caused by earthquakes in the Atsuma River basin using Landset satellite images; (ii) to assess changes in the

hydrological processes in the Atsuma River basin after the earthquake; and (iii) to predict the future conditions of the hydrological processes based on human governance activities.

## 2. Data and Methods

### 2.1. Study Ragion

The Atsuma River basin is located in southern Hokkaido between latitudes 42°34′ and 42°53′ N and between longitudes 142°7′ and 141°59′ E. It serves an area of approximately 382.9 km$^2$, covers a stretch of 52.3 km, and has an altitude range of 0–640 m above sea level. Rainfall, snowmelt water, and base flow are the most important water sources in the river. In the last two decades, the Atsuma River basin has experienced annual average precipitation of 1235.9 mm, average wind speed of 2.5 m/s, average solar radiation of 8 MJ/m$^2$, average relative humidity of 78%, minimum and maximum temperatures of −25.3 and 32.3 °C, respectively, and an annual average temperature of 6.6 °C. Figure 1 shows the Atsuma River basin.

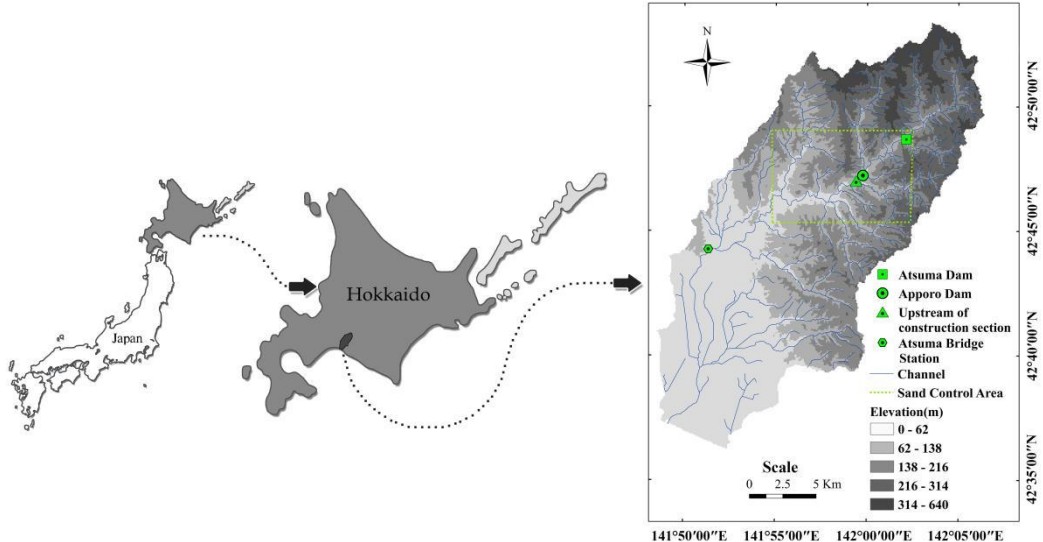

**Figure 1.** The Atsuma River basin.

### 2.2. Main Event

The epicenter of the 2018 Hokkaido Eastern Iburi earthquake was at 42.7° N, 142° E, with a depth of 37 km. In Atsuma, an Mj 6.7 (Mw 6.6) earthquake with a maximum seismic intensity of 7 was observed (according to the Japan Meteorological Agency). It was the first earthquake in Hokkaido, Japan, of such a large seismic intensity. After the earthquake, there were several large-scale deep-seated landslides and multiple shallow landslides over approximately 400 km$^2$ of hilly areas with elevations of 200–400 m.

In terms of human and property damage, the earthquake killed 41 people in Hokkaido and injured at least 692, including 13 serious injuries and 679 minor injuries. The earthquake also damaged at least 2508 buildings and several roads, and buried several cars in mudslides. In terms of production and daily life, the earthquake caused damage to equipment at the Tomato Atsuma Electric Power Station, the largest thermal power plant in Hokkaido, and Unit 2 of the Onbetsu Electric Power Station in Ibetsu, Kushiro. Hokkaido lost more than half of its power supply, resulting in a total of 2.95 million households across Hokkaido being without electricity. Running water and telecommunications were also suspended in some parts of Hokkaido, supermarkets and convenience stores were in short supply, and Hokkaido's 1800 schools were temporarily closed. Hokkaido's agriculture, forestry, aquaculture, and aquaculture experienced serious economic losses because of the power outages and geological disaster that were caused by earthquake. Several Hokkaido plants, including SUMCO Chitosa fabs, CALBEE and Sapporo Breweries, have suspended production.

In the Atsuma River basin, five main rainfall events occurred from April 2018 to August 2018. Some Studies have shown that continuous rainfall and strong motion were main contributors to the failure of the regional slopes [13]. The basement complex in the affected area (which consists of sedimentary rocks) was covered with thick pyroclastic rocks, and the strong seismic shocks triggered shallow landslides, which moved along valley-type topography instead of the planar slope topography and traveled greater distances. Some shallow landslides occurred on relatively gentle slopes (<30°). Furthermore, some studies have shown that, after the earthquakes and its landslides, the safety factors under natural conditions in the Atsuma River basin were such that no slopes would slide when there was no rainfall [13]. Therefore, it can be inferred that the main source of sediment in the Atsuma River basin after the earthquake was erosion. Figure 2a,b show the slope degree map and slope direction map of the Atsuma River basin, respectively.

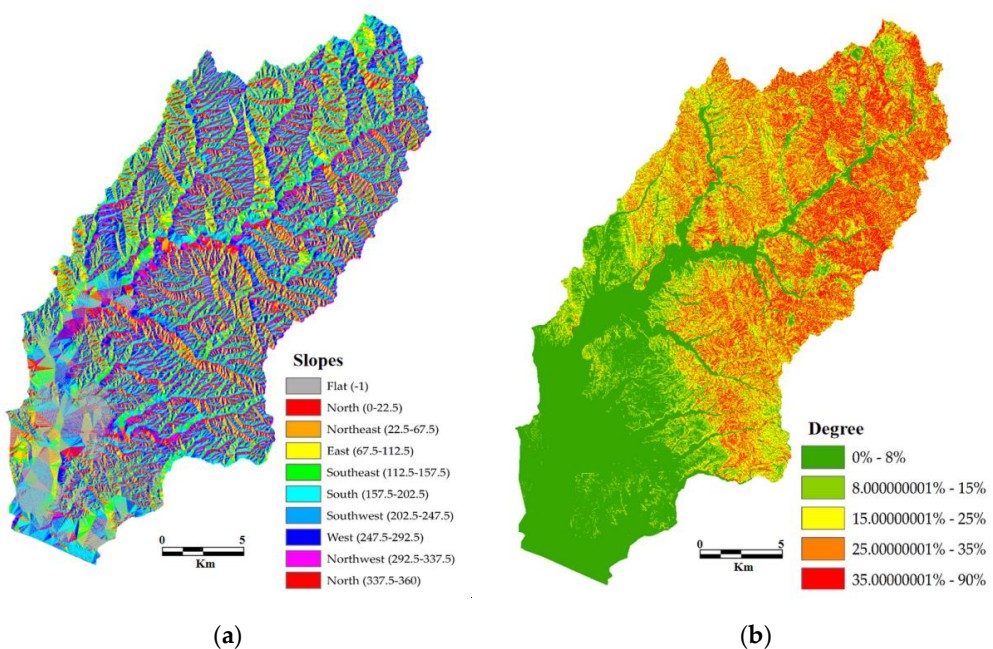

(**a**)　　　　　　　　　　　　　　　　　　　　　（**b**)

**Figure 2.** (**a**) Slope degree map [22] and (**b**) slope direction map of the Atsuma River basin.

The Atsuma River basin area includes the Apporo and Atsuma dams. The body of the Atsuma dam was damaged during the Hokkaido Eastern Iburi earthquake. The drainage channel was blocked due to dirt, sand, trees, and other debris, which made the dam unusable. However, debris removal and restoration projects have begun, and the Atsuma Dam should be restored for use in 2023. While the body of the Apporo Dam was not damaged, large-scale landslides occurred in the mountains and streams around the dam. As a result, large amounts of sediment and driftwood accumulated in the dam and the surrounding slopes, which severely impacted the water storage function of the dam. This function of the dam will be restored by removing the accumulated sediment and driftwood, cleaning it up, and reinforcing the unstable slopes. Furthermore, after the earthquake, large-scale mountain collapses occurred in the Hidaka-horonai, Chichepe, Chicaep, and Towa rivers, which blocked the river channel. Therefore, the slope will be reinforced and sand control dams will be built in these areas. Figure 3 shows a sand control map of the Atsuma River basin.

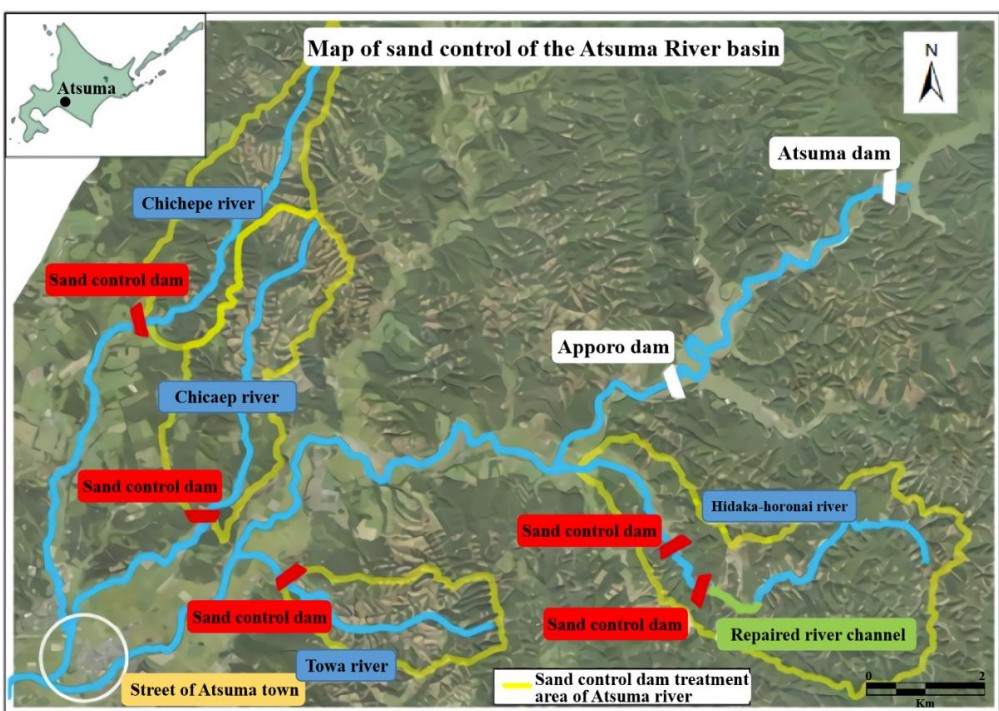

**Figure 3.** Sand control map of the Atsuma River basin.

### 2.3. Hydrological Modeling

The SWAT model is a physically based hydrologic model, developed by the United States Department of Agriculture (USDA). It is used to simulate the quality and quantity of surface and ground water and predict the environmental impact of LULC, land management practices, and climate change.

Equation (1), which is used by the SWAT model to simulate runoff, is as follows:

$$SW_t = SW_o + t\sum_{i=1} \left( R_{day} - Q_{surf} - E_a - W_{seep} - Q_{gw} \right) \tag{1}$$

where $SW_t$ is final soil moisture content (mm), $SW_o$ is initial water content (mm), t is the time (days), $R_{day}$ is the precipitation on day i (mm), $Q_{surf}$ is the surface runoff on day i (mm), $E_a$ is the evapotranspiration on day i (mm), $W_{seep}$ is the amount of water seeping into the soil profile (mm), and $Q_{gw}$ is the amount of return flow on day i (mm).

Equation (2), which is used by SWAT model to simulate the erosion and sediment transport, is as follows:

$$Sed = 11.8 \times \left( Q_{surf} \times q_{peak} \times area_{hru} \right)^{0.56} \times K \times C \times P \times LS \times CFRG \tag{2}$$

where Sed is the sediment yield (tons/day) of a hydrological response unit (HRU), $Q_{surf}$ is the volume of surface runoff (mm/$10^3 \cdot m^2$), $q_{peak}$ is the peak runoff rate ($m^3$/s), $area_{hru}$ is the HRU area ($10^3 \cdot m^2$), K is the universal soil loss equation (USLE) soil erodibility factor (dimensionless), C is the USLE cover and management factor (dimensionless), P is the USLE support practice factor (dimensionless), LS is the USLE topographic factor (dimensionless), and CFRG is the coarse fragment factor (dimensionless).

In order to modify the parameters and improve the simulation results, the SWAT calibration and uncertainty program (SWAT-CUP) is used to calibrate and validate the SWAT model and perform sensitivity (one-at-a-time and global) and uncertainty analyses. The uncertainty analysis method used in this study is the sequential uncertainty fitting procedure version 2 (SUFI-2), which is part of the SWAT-CUP platform. SUFI-2, based on a Bayesian framework, operates within the uncertainty domains (prior and posterior) that

are associated with each parameter. The uncertainties surrounding the parameters mainly include three aspects: the input datasets, the model structure, and the measured data. Further, SUFI-2 performs combined optimization and uncertainty analysis using a global search procedure and deals with several parameters through Latin hypercube sampling.

### 2.4. Datasets

We used meteorological data (precipitation, temperature, average wind speed, relative humidity, and solar radiation) for 2009–2020, taken from the Japan Meteorological Agency and the Japan Meteorological Business Support Center, and river observation data (suspended solids, runoff, and inflow and outflow data of dams) for 2015–2020, taken from the Hokkaido Government Iburi General Sub-Prefectural Bureau, Muroran Construction Management Department (Supplementary Materials). However, the data is missing for some time periods. Table 1 shows the datasets used in this study. Table S10 shows the information of dataset (Supplementary Materials).

In this study, we used ENVI soft to process the Landsat-8 image and develop the LULC maps. Six LULC classes, including farmland, forest, grass, water, bare land, and building land, were identified and classified, as listed in Table 2. Figure 4 shows the 2015 and 2020 LULC conditions in the Atsuma River basin. To simulate the SWAT model, we also used DEM and soil data, as shown in Table 1.

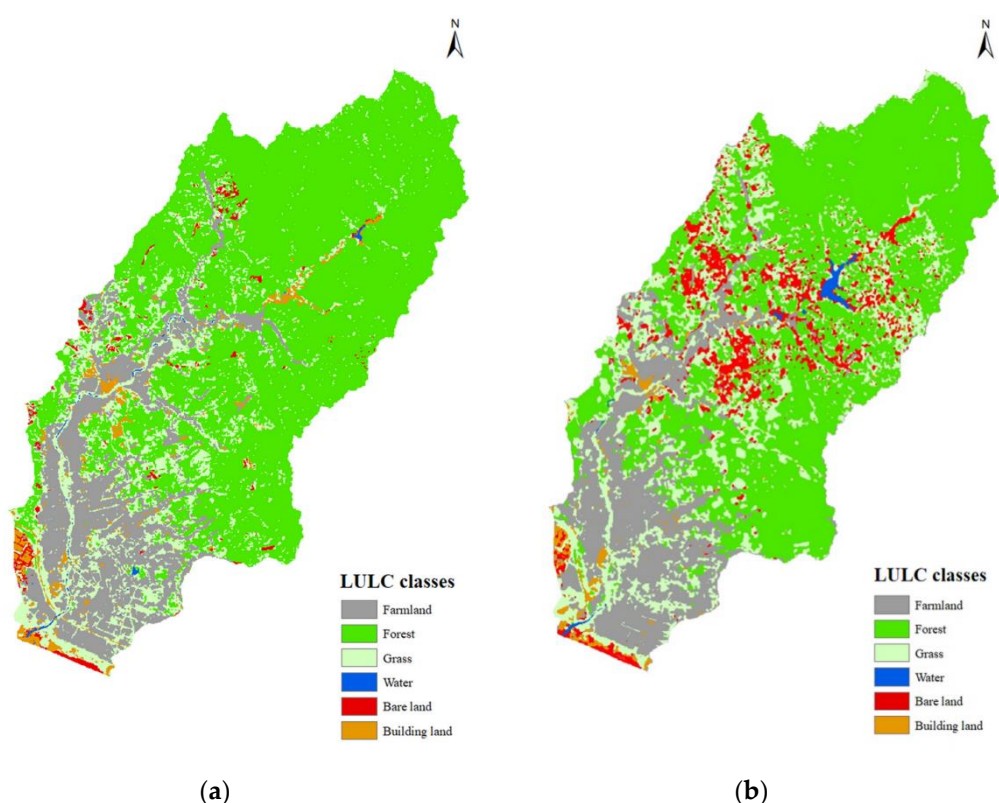

**Figure 4.** LULC conditions in the Atsuma River basin for (**a**) 2015 and (**b**) 2020.

**Table 1.** Dataset in this study.

| Data Type | Description | Resolution | Source | Download Path | Data number | Acquisition date | Formats |
|---|---|---|---|---|---|---|---|
| Topography map | Digital elevation model (DEM) | 0.4″ ×0.4″ (about 10 m square) | Geographical information Authority of Japan | https://fgd.gsi.go.jp/download/mapGis.php?tab=dem (accessed on 20 November 2021) | DEM10B 6342 | 3 November 2019 | TIF |
| Land use and land cover map | Land use and land cover classifications | 30 m (OLI) 100 m (TIRS) | United States Geological Survey | https://earthexplorer.usgs.gov/ (accessed on 20 November 2021) | LC08_L2SP_107030_20150923_20200908_02, LC08_L2SP_107030_20200531_20200820_02 | 15 March 2021 | |
| Soils map | Soil types | 1/200,000 | Ministry of Land, Infrastructure, Transport and Tourism of Japan | https://nlftp.mlit.go.jp/kokjo/inspect/landclassification/land/l_national_map_20-1.html (accessed on 20 November 2021) | Hokkaido_1 200,000 Land Classification Basic Survey | 3 November 2019 | GRID |
| Meteorological data | Radar precipitation data | Daily (450 stations) | Japan Meteorological Business Support Center | http://www.jmbsc.or.jp/jp/ (accessed on 20 November 2021) | 1 January 2009–31 December 2020 | 9 May 2021 | CSV |
| | Minimum and maximum temperature Wind speed | Daily (Atsuma station) | Japan Meteorological Agency | https://www.data.jma.go.jp/gmd/risk/obsdl/index.php(accessed on 20 November 2021) | | | |
| | Relative humidity | Daily (Tomakomai station) | | | | | |
| | Solar radiation | Daily (Sapporo station) | | | | | TXT |

**Table 2.** LULC distribution in the Atsuma River basin in 2015 and 2020 (km$^2$).

|  | LULC | 2015 | 2020 |
|---|---|---|---|
| 1 | Farmland | 64.06 | 67.36 |
| 2 | Forest | 232.25 | 186.35 |
| 3 | Grass | 55.5 | 78.26 |
| 4 | Water | 0.88 | 18.71 |
| 5 | Bare land | 4.48 | 25.46 |
| 6 | Building land | 9.69 | 7.7 |

*2.5. Methods*

The following procedure was followed for the quantification of LULC change and its impact on the runoff and sediment transport processes in the Atsuma River basin: (i) the 2015 and 2020 LULC maps of the Atsuma River basin were developed to study the rapid and drastic change in LULC caused by earthquakes in the river basin; (ii) the SWAT model was used to simulate the runoff and sediment transport processes before and after the earthquake based on LULC observations from 2015 and 2020 to analyze the changes in the runoff and sediment transport processes after the earthquake; (iii) based on governance activity, the future conditions of runoff and sediment transport in the Atsuma River basin were predicted. Figure 5 shows the technology roadmap of this study.

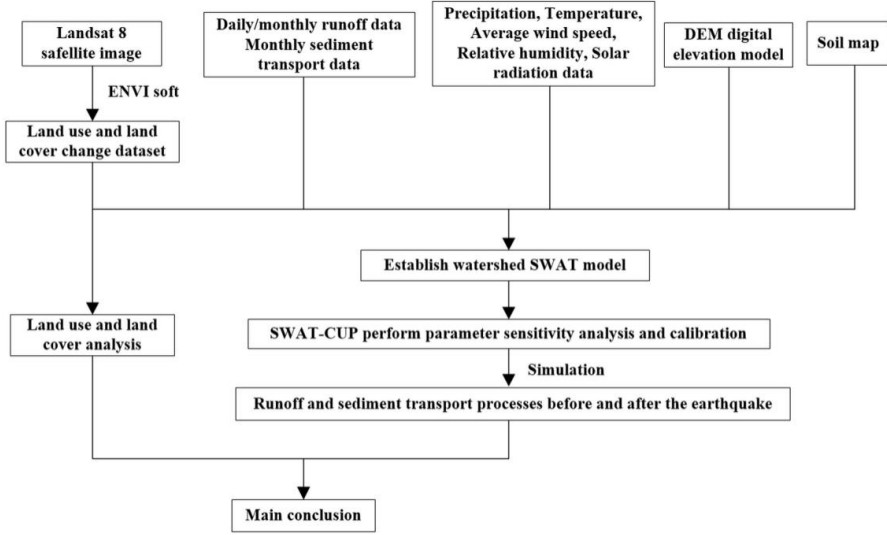

**Figure 5.** Technology roadmap of this study.

The simulation results were evaluated using the coefficient of determination (R$^2$), the Nash–Sutcliffe efficiency coefficient (NSE), percent bias (PBIAS), and the Pearson product-moment correlation coefficient (PPMCC). The calculation process formulas of R$^2$, NSE, PBIAS, and PPMCC are shown in Equations (3)–(6):

$$R^2 = \frac{\sum\limits_{i=1}^{T}\left(O_i - \bar{O}\right)\left(S_i - \bar{S}\right)}{\sqrt{\sum\limits_{i=1}^{T}\left(O_i - \bar{O}\right)^2 \sum\limits_{i=1}^{T}\left(S_i - \bar{S}\right)^2}} \tag{3}$$

$$NSE = 1 - \frac{\sum\limits_{i=1}^{T}(O_i - S_i)^2}{\sum\limits_{i=1}^{T}\left(O_i - \bar{O}\right)^2} \tag{4}$$

$$\text{PBIAS} = \frac{\sum\limits_{i=1}^{T} (O_i - S_i)}{\sum\limits_{i=1}^{T} O_i} \tag{5}$$

$$\text{PPMCC} = \frac{\sum\limits_{i=1}^{T} \left( O_i - \bar{O} \right) \left( S_i - \bar{S} \right)}{\sqrt{\sum\limits_{i=1}^{T} \left( O_i - \bar{O} \right)^2 \sum\limits_{i=1}^{T} \left( S_i - \bar{S} \right)^2}} \tag{6}$$

where T is the calculation time (days/months), $O_i$ is the observation runoff at time i, $S_i$ is the calculation runoff at time i, $\bar{O}$ is the observation of the average runoff, and $\bar{S}$ is the calculated average runoff.

The performance of a model is considered good when NSE > 0.6, $R^2$ > 0.7, PBIAS = ±15%, and PPMCC > 0.8.

## 3. Results and Discussion

### 3.1. Land Use and Land Cover Change Analysis

On comparing the 2015 and 2020 land use maps of the Atsuma River basin, it was found that the farmland area decreased by 3.3 km$^2$ and remained stable with no major changes. The forest area and building area decreased significantly by 45.9 km$^2$ and 1.99 km$^2$, respectively. The bare land area, grassland area, and water area increased significantly by 20.98 km$^2$, 22.75 km$^2$, and 17.83 km$^2$, respectively. Through the analysis, it was found that most of the decrease in the forest area was connected to the increases in new bare land and grassland, which can be attributed to the 2018 Hokkaido Eastern Iburi earthquake and the landslides that it caused. The Apporo dam area was transformed from building land into a water body owing to the completion of the Apporo Dam in 2017. However, because the Atsuma Dam was damaged and not functional after the earthquake, the Atsuma dam area was transformed from water to bare land. Figure 6 compares the slope collapse and sediment accumulation area after the earthquake with the bare land and the grassland area in the 2020 LULC Atsuma River basin map.

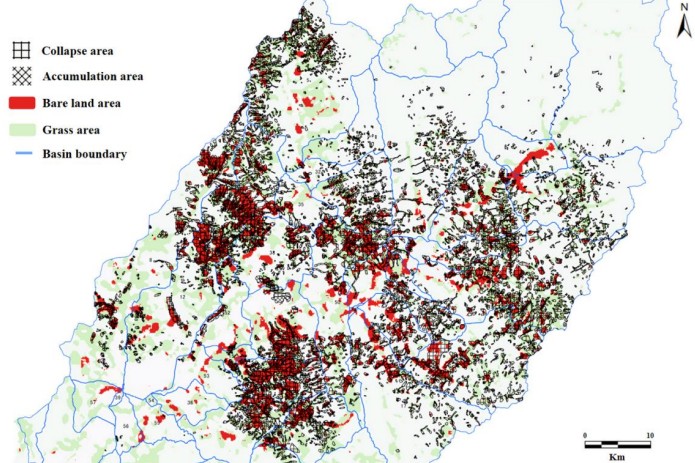

**Figure 6.** Comparison map of the slope collapse, sediment accumulation area and bare land, and grass area in the Atsuma River basin.

The Atsuma Dam is expected to be restored for use in 2023, which means that the Atsuma dam area will be transformed from bare land into water. Furthermore, the construction of the reinforced slopes and sand control dams of the Hidaka-horonai, Chichepe, Chicaep, and Towa rivers in the basin has been completed. Among these, the Hidaka-horonai river area is 63.17 km$^2$, and the Chichepe river area is 43.2 km$^2$. The sand control dams mainly arrest river sediment, regulate sediment transportation, prevent erosion,

control the flow center, and restrain soil and rock flow. Although the construction of sand control dams does not affect the LULC condition, they significantly affect the amount of sediment transport in the basin. Currently, there is no construction plan for large-scale vegetation restoration or slope reinforcement on new bare land. Therefore, in the short-term, the LULC condition in the basin is not expected to change drastically, and the runoff in the basin will not change significantly. However, the amount of sediment transport in the basin will decrease significantly owing to the construction of the sand control dam.

### 3.2. Runoff

The SWAT model can generate default parameters based on the input of geographic data. However, often it cannot accurately simulate the runoff. In this study, the runoff parameters of the SWAT model were adjusted using SWAT-CUP. First, the most sensitive parameters for simulating runoff were determined by the global sensitivity part of SWAT-CUP. Second, the sequential uncertainty fitting (SUFI-2) algorithm was used to obtain the corrected runoff parameters (Table 3). Herein, the warm-up, calibration, and validation periods were from 2009 to 2014, 2015 to 2017, and 2018 to 2020, respectively.

**Table 3.** Runoff parameters.

| No | Name | Description | Daily | | | | Monthly | | | |
|---|---|---|---|---|---|---|---|---|---|---|
| | | | 2015 | 2020 | 2015 | 2020 | 2015 | 2020 | 2015 | 2020 |
| | | | Default Values | | Corrected Values | | Default Values | | Corrected Values | |
| 1 | TIMP.bsn | Snowpack temperature lag factor | 1 | 1 | 1 | | 1 | 1 | 0.161241 | |
| 2 | CN2.mgt | Initial SCS runoff curve number for moisture condition II | 79 | 87 | 63.189432 | 69.588362 | 79 | 87 | 46.119094 | 50.789382 |
| 3 | SMFMN.bsn | Melt factor for snow on December 21 (mm $H_2O/°C$-day) | 4.5 | 4.5 | 1.359352 | | 4.5 | 4.5 | 4.916194 | |
| 4 | SMFMX.bsn | Melt factor for snow on June 21 (mm $H_2O/°C$-day) | 4.5 | 4.5 | 2.048258 | | 4.5 | 4.5 | 1.638298 | |
| 5 | TLAPS.sub | Temperature lapse rate (°C/km) | 0 | 0 | −2.392 | | 0 | 0 | −1.221248 | |
| 6 | SOL_AWC.sol | Available water capacity of the soil layer (mm $H_2O$/mm soil) | 0.1 | 0.1 | 0.025737 | | 0.1 | 0.1 | 0.09 | |
| 7 | SMTMP.bsn | Snowmelt base temperature (°C) | 0.5 | 0.5 | 0.108525 | | 0.5 | 0.5 | 3.543045 | |
| 8 | CH_K2.rte | Effective hydraulic conductivity in tributary channel alluvium (mm/hrh) | 0 | 0 | 52.697681 | | 0 | 0 | 16.700487 | |
| 9 | CH_K1.sub | Effective hydraulic conductivity in tributary channel alluvium (mm/h) | 0 | 0 | 12.198375 | | 0 | 0 | 0.503748 | |
| 10 | SNOCOVMX.bsn | Minimum snow water content that corresponds to 100% snow cover, SNO100 (mm $H_2O$) | 1 | 1 | 0.639671 | | 1 | 1 | 0.375468 | |
| 11 | CH_N2.rte | Manning's "*n*" value for the main channel | 0.014 | 0.014 | 0.304582 | | 0.014 | 0.014 | 0.032119 | |
| 12 | CH_N1.sub | Manning's "*n*" value for the main channel | 0.014 | 0.014 | 0.230863 | | 0.014 | 0.014 | 0.513274 | |
| 13 | ESCO.hru | Soil evaporation compensation factor | 0.95 | 0.95 | 0.7684 | | 0.95 | 0.95 | 0.61147 | |
| 14 | PLAPS.sub | Precipitation lapse rate (mm $H_2O$/km) | 0 | 0 | 173.700012 | | 0 | 0 | −109.683922 | |
| 15 | SFTMP.bsn | Snowfall temperature (°C) | 1 | 1 | −0.908274 | | 1 | 1 | −1.100187 | |
| 16 | ALPHA_BF.gw | Baseflow alpha factor (days) | 0.048 | 0.048 | 0.972874 | | 0.048 | 0.048 | 0.004363 | |
| 17 | SURLAG.bsn | Surface runoff lag coefficient | 4 | 4 | 4 | | 4 | 4 | 0.932302 | |
| 18 | GW_DELAY.gw | Groundwater delay time (days) | 31 | 31 | 62.261829 | | 31 | 31 | 84.258835 | |
| 19 | SNO50COV.bsn | Fraction of snow volume represented by SNOCOVMX that corresponds to 50% snow cover. | 0.5 | 0.5 | 0.743382 | | 0.5 | 0.5 | 0.771502 | |
| 20 | GWQMN.gw | Threshold depth of water in the shallow aquifer required for return flow to occur (mm $H_2O$). | 1000 | 1000 | 0.362763 | | 1000 | 1000 | 2.610365 | |

To study the impact of LULC change on the runoff process, we first simulated the daily/monthly runoff process in the Atsuma River basin based on the 2015 LULC data. Next, the calibrated SWAT model and the 2020 LULC data were used to simulate the daily/monthly runoff process in the Atsuma River basin, without changing the soil, elevation, and meteorological data. Figures 7–10 show the runoff simulation results. Table 4 shows the error analysis results. For the 2015 LULC data, the daily runoff simulation results in the calibration period had an $R^2$ of 0.876, NSE of 0.707, PBIAS of 10.52%, and PPMCC of 0.876. Similarly, for the monthly runoff simulation results, $R^2$ was 0.910, NSE was 0.691, PBIAS was 16.15%, and PPMCC was 0.911. For the daily runoff simulation results in the verification period, $R^2$ was 0.874, NSE was 0.620, PBIAS was −6.85%, and PPMCC was 0.874. For the monthly runoff simulation results, $R^2$ was 0.861, NSE was 0.729, PBIAS was −6.028%, and PPMCC was 0.610. As a result, the runoff simulation error was small and the simulation effect was good, indicating that the SWAT model can accurately simulate the runoff process in the Atsuma River basin.

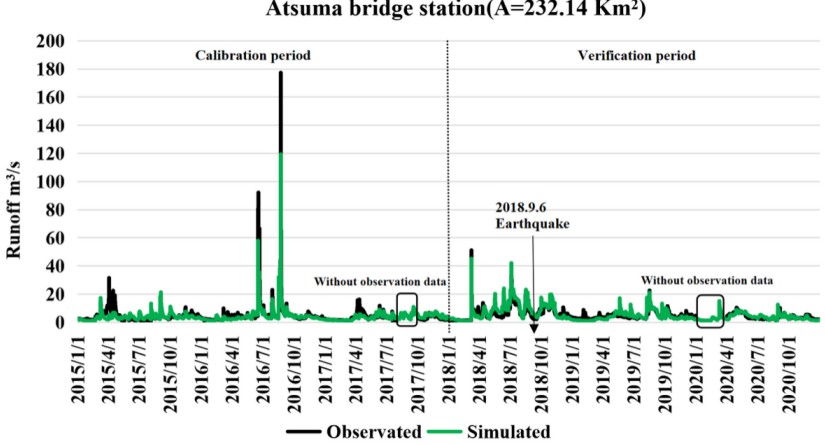

**Figure 7.** Daily simulated runoff calculated from the 2015 LULC data.

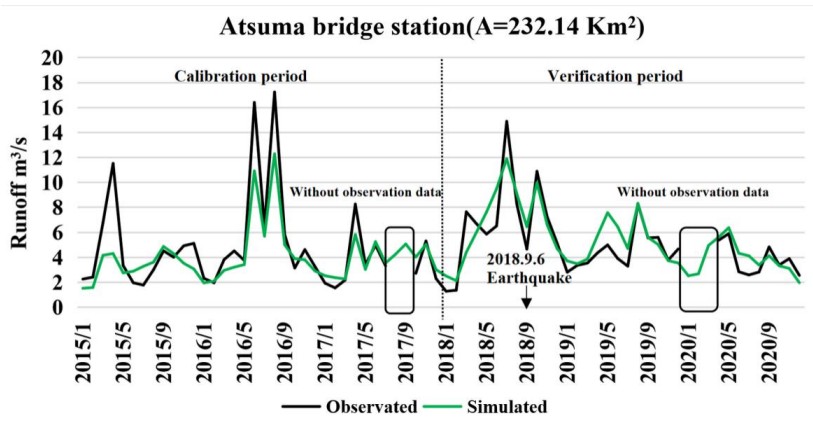

**Figure 8.** Monthly simulated runoff calculated from the 2015 LULC data.

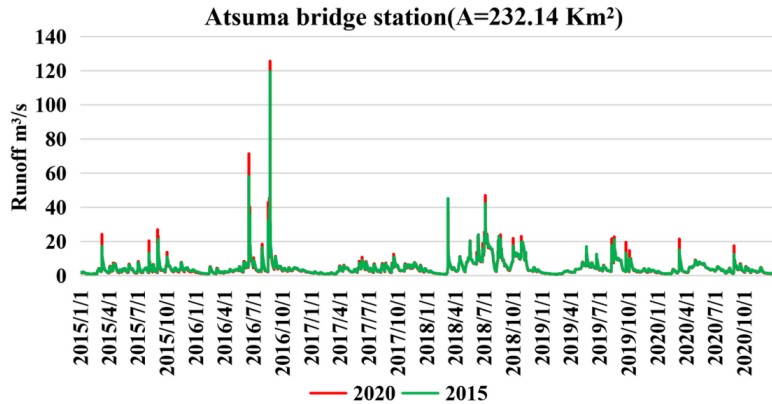

**Figure 9.** Comparison of simulated daily runoff based on the 2015 and 2020 LULC data.

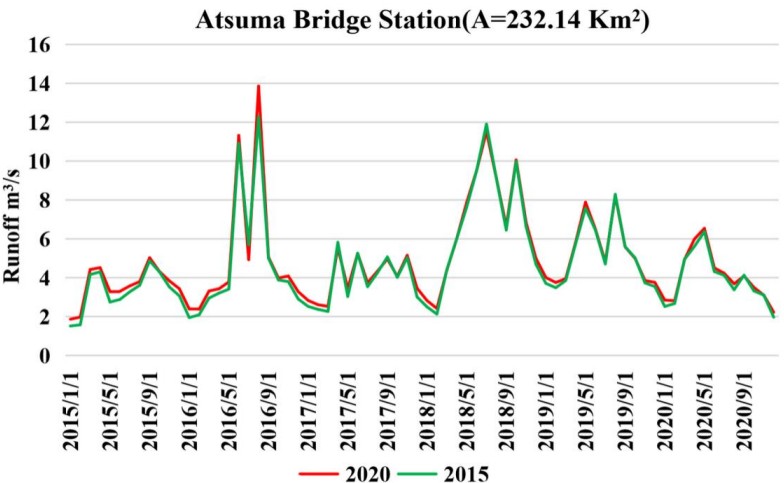

**Figure 10.** Comparison of simulated monthly runoff based on the 2015 and 2020 LULC data.

**Table 4.** Error analysis of the SWAT model runoff simulation based on the 2015 LULC data.

|  | Atsuma Bridge Station | $R^2$ | NSE | PBIAS | PPMCC |
|---|---|---|---|---|---|
| Daily | Calibration period (2015–2017) | 0.876 | 0.707 | 10.52% | 0.876 |
|  | Verification period (2018–2020) | 0.874 | 0.620 | −6.85% | 0.874 |
| Monthly | Calibration period (2015–2017) | 0.910 | 0.691 | 16.15% | 0.911 |
|  | Verification period (2018–2020) | 0.861 | 0.729 | −6.028% | 0.610 |

By comparing the runoff simulation results obtained using the 2015 and 2020 LULC data, it was seen that the simulation results were very similar and the curve was consistent, indicating that the LULC change in the Atsuma River basin from 2015 to 2020 has had little impact on the runoff process. However, after careful observation, it was found that the peak of the daily/monthly runoff simulation results obtained for the 2020 data was slightly larger than that of 2015. This indicates that the decrease in forest land area and the increase in the bare land area slightly increased the proportion of precipitation that is transformed into runoff in the Atsuma River basin. By analyzing the water budget of the Atsuma River basin from 2015 to 2020 (Table 5), it was found that the proportion of precipitation that is transformed into runoff was higher from 2018 to 2020 compared to 2015 to 2017, thereby showing an increasing trend, which accompanied the LULC change from 2015 to 2020.

**Table 5.** Upstream water budget of the Atsuma Bridge station from 2015 to 2020 (m$^3$/a).

| Year | 2015 | 2016 | 2017 | 2018 | 2019 | 2020 | Average |
|---|---|---|---|---|---|---|---|
| Precipitation | 1274 | 1540 | 1171 | 1451 | 1067 | 985 | 1248 |
| Rainfall | 994 | 1299 | 1068 | 1253 | 999 | 915 | 1088 |
| Snowfall | 280 | 241 | 103 | 198 | 68 | 70 | 160 |
| Runoff | 585 | 835 | 490 | 918 | 615 | 518 | 660 |
| Runoff/Precipitation | 0.46 | 0.54 | 0.42 | 0.63 | 0.58 | 0.53 | 0.53 |

### 3.3. Sediment Transport

Keeping the runoff parameters the same, the sediment-sensitive parameters were determined using SWAT-CUP, and the corrected sediment parameters were obtained using the SUFI-2 algorithm (Table 6). The 2015 and 2020 LULC data were used to simulate the sediment transport process before (January 2017–August 2018) and after the earthquake (October 2018–December 2020), respectively. Figures 11 and 12 show the simulation results of the sediment transport process, and the error analysis results are presented in Table 7. The simulation error measures for the sediment transport process before the earthquake were $R^2$ = 0.802, NSE = 0.621, PBIAS = 10.205% and PPMCC = 0.802. The simulation error measures for the sediment transport process after the earthquake were $R^2$ = 0.823, NSE = 0.638, PBIAS = 15.755%, and PPMCC = 0.823. The results show that the simulation error of the sediment transport process before and after the earthquake was small and that the simulation effect was good, indicating that the SWAT model can successfully and accurately estimate the sediment transport process of the Atsuma River basin.

**Table 6.** Sediment parameters.

| No | Name | Description | Before Earthquake | | After Earthquake | |
|---|---|---|---|---|---|---|
| | | | Default Values | Corrected Values | Default Values | Corrected Values |
| 1 | USLE_P.mgt | USLE equation support practice factor | 1 | 0.090052 | 1 | 0.060287 |
| 2 | SOL_BD().sol | Moist bulk density (Mg/m$^3$ or g/cm$^3$) | 0.3 | 1.37037 | 0.3 | 1.954686 |
| 3 | REVAPMN.gw | Threshold depth of water in the shallow aquifer for "revap" or percolation to the deep aquifer to occur (mm H$_2$O) | 750 | 186.166534 | 750 | 401.453583 |
| 4 | USLE_C{2}.plant.dat | Minimum value of USLE C factor for water erosion applicable to the land cover/plant (Forest) | 0.001 | 0.0008159 | 0.001 | 0.00102348 |
| 5 | SPCON.bsn | Linear parameter for calculating the maximum amount of sediment that can be reentrained during channel sediment routing | 0.0001 | 0.006331 | 0.0001 | 0.002783 |
| 6 | USLE_C{3}.plant.dat | Minimum value of USLE C factor for water erosion applicable to the land cover/plant (Grass) | 0.003 | 0.00083717 | 0.003 | 0.00069073 |
| 7 | CH_ERODMO.rte | CH_ERODMO is set to a value between 0.0 and 1.0. A value of 0.0 indicates a non-erosive channel, while a value of 1.0 indicates no resistance to erosion. | 0 | 3.744806 | 0 | 8.263742 |
| 8 | HRU_SLP.hru | Average slope steepness (m/m) | 0.05155 | 0.0708287 | 0.6687 | 0.58217958 |
| 9 | USLE_K.sol | USLE equation soil erodibility (K) factor (units: 0.013 (metric ton m$^2$ hr)/(m$^3$-metric ton cm)) | 0.0628 | 0.3016999 | 0.0628 | 0.484151 |
| 10 | EPCO.hru | Plant uptake compensation factor | 1 | 0.490527 | 1 | 0.992019 |
| 11 | SOL_K.sol | Saturated hydraulic conductivity (mm/hr) | 10.63 | 6.69041 | 10.63 | 13.267877 |
| 12 | OV_N.hru | Manning's "n" value for overland flow | 0.1 | 0.0937598 | 0.14 | 0.249963 |
| 13 | GW_REVAP.gw | Groundwater "revap" coefficient | 0.02 | 0.02342478 | 0.02 | 0.01479886 |
| 14 | CH_COV1.rte | Channel cover factor | 0 | 25.236423 | 0 | 11.92763 |
| 15 | CH_L2.rte | Length of main channel (km) | 10.9016 | 10.4795119 | 10.9016 | 13.864849 |
| 16 | CH_S2.rte | Average slope of tributary channels (m/m) | 0.011634 | 0.0117971 | 0.011634 | 0.02539592 |

**Table 6.** *Cont.*

| No | Name | Description | Before Earthquake | | After Earthquake | |
|---|---|---|---|---|---|---|
| | | | Default Values | Corrected Values | Default Values | Corrected Values |
| 17 | CH_COV2.rte | Channel cover factor | 0 | 22.798399 | 0 | 21.584068 |
| 18 | USLE_C{1}.plant.dat | Minimum value of USLE C factor for water erosion applicable to the land cover/plant (Farmland) | 0.001 | 0.0013788 | 0.001 | 0.00027133 |
| 19 | SPEXP.bsn | Exponent parameter for calculating sediment reentrained in channel sediment routing | 1 | 1.321381 | 1 | 1.156308 |
| 20 | SLSUBBSN.hru | Average slope length (m) | 60.97561 | 60.97561 | 9.14634 | 7.10126411 |

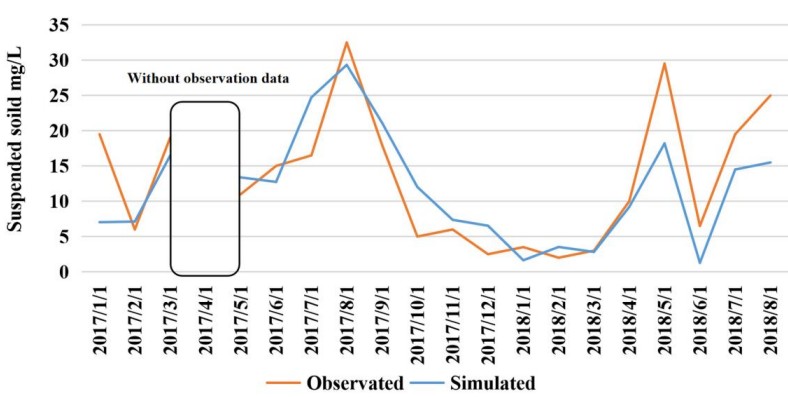

**Figure 11.** Sediment transport simulation based on the 2015 LULC data.

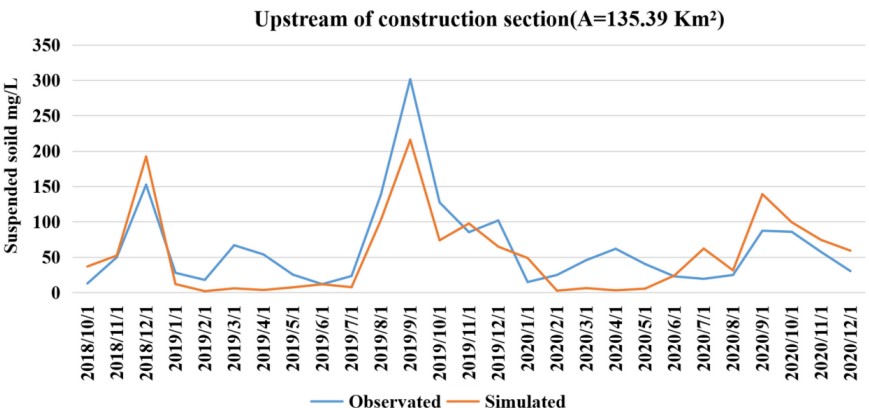

**Figure 12.** Sediment transport simulation based on the 2020 LULC data.

**Table 7.** Error analysis of the SWAT model for the sediment transport simulation.

| Upstream of Construction Section | $R^2$ | NSE | PBIAS | PPMCC |
|---|---|---|---|---|
| January 2017–August 2018 | 0.802 | 0.621 | 10.205% | 0.802 |
| October 2018–December 2020 | 0.823 | 0.638 | 15.755% | 0.823 |

By comparing the simulation results in Figures 11 and 12, it can be seen that the sediment transport after the earthquake increased by an order of magnitude compared to that before the earthquake. This indicates that the sediment transport in the Atsuma River basin increased significantly after the earthquake. Figure 13 shows the accumulation of transported sediment, adjusted according to the runoff, before and after the earthquake. First, the rate of sediment transportation increased significantly after the earthquake, becoming 4.42 times greater than it was before the earthquake. Second, by analyzing

the cumulative curve before the earthquake, it was found that the sediment transport rate was relatively stable, indicating that soil erosion and sediment transport capacity in the Atsuma River basin was relatively stable before the rapid LULC change that was caused by the earthquake and landslide. Moreover, by analyzing the cumulative curve after the earthquake, it was found that the sediment transport rate increased significantly and became unstable after the earthquake. The cumulative curve after the earthquake can be roughly divided into the following six stages: 6 September–October 2018, November–December 2018, January–July 2019, August–December 2019, January–June 2020, and July–December 2020.

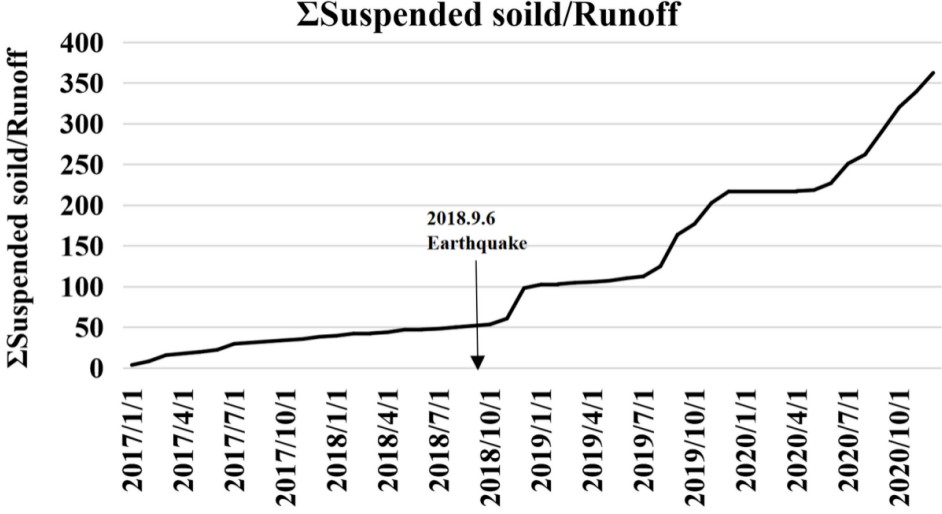

**Figure 13.** Accumulation of transported sediment, adjusted according to runoff.

From 6 September 2018 to October 2018, the rate of sediment transportation did not immediately increase owing to the damage caused to the Apporo dam by the earthquake, because this decreased the runoff and the capacity to transport sediment. Therefore, the rate of sediment transportation increased minimally. From November to December 2018, the runoff increased with the release of water following construction, and the flow transported the sediment that had accumulated in and around the river channel to downstream of the river. Therefore, the sediment transport rate increased significantly. From January 2019 to July 2019, first, because the precipitation was in the form of snowfall from January to March 2019 in the Atsuma River basin and the temperature was low, the snowfall mainly existed in the form of snow cover and did not transform into runoff. Second, from April 2019 to July 2019, the runoff was small owing to the construction of the Apporo dam, and the capacity to transport sediment was weak, resulting in a lower sediment transport rate. From August 2019 to December 2019, when the Apporo dam was functional, the water levels rose in the Atsuma River, and the runoff transported the sediment to downstream of the river. Furthermore, the sediment that was produced under the erosion of flow was transported by runoff to downstream of the river. Therefore, the rate of sediment transport increased significantly at this stage. From January 2020 to June 2020, also because the precipitation was mainly in the form of snow, snowfall mainly existed in the form of snow cover. Moreover, there is very little rain in spring, and so the runoff was mainly due to the reservoir's seasonal outflow, which is part of the river's continual flow. At this time, the basin erosion was mainly caused by river channel erosion. Therefore, the change in the rate of sediment transportation was less in this stage—in fact, it was basically stable. From July 2020 to December 2020, with the advent of the rainy season, the erosion process of the basin was mainly due to the erosion of soil and river channels that occurred as a result of precipitation, confluence, runoff, and other processes. Owing to the significant increase in bare land in the basin after the earthquake, the erosion effect strengthened, and more sediment was generated in the river channel and transported downstream by the runoff.

Therefore, the rate of sediment transport increased significantly at this stage. In general, the amount of sediment transported increased significantly after the earthquake.

## 4. Conclusions

This study evaluated the abrupt changes in land use and land cover after the occurrence of the Hokkaido Eastern Iburi earthquake in 2018, and its impact on the runoff and sediment transport processes of the Atsuma River basin. The object of the research was to quantify the LULC change in the Atsuma River basin, and the SWAT model was used to evaluate how LULC change affected the hydrological process of the Atsuma River basin to predict the future situation of the runoff and sediment transport processes based on human repairing activities. In addition, the research ideas and methods of this study are applicable to different scenarios in similar situations, but the conclusions of this study are only applicable to the impact of 2018 Hokkaido Eastern Iburi earthquake on the Atsuma River basin. The conclusions of this study are as follows: (i) the abrupt change in land use caused by the earthquake in the Atsuma River basin mainly meant that nearly 10% of the forest area was transformed into bare land; (ii) although the abrupt LULC change caused a slight increase in the runoff, it significantly increased the sediment transport rate by approximately 3.42 times in the Atsuma River basin, compared to before the earthquake; and (iii) although the runoff situation of the Atsuma River basin is not expected to change significantly in the future, the amount of sediment carried across the river basin can be reduced by the active governance activities of humans. It can be seen from this case that strong earthquakes have a greater impact on land use and land cover in river basins, and changes in land use and land cover can affect the hydrological processes of the basins.

In this study, which was based on limited observational data, the LULC change after the earthquake was quantified, the impact of LULC change after the earthquakes on the runoff and sediment transport in the Atsuma River basin was verified, but the long-term impacts need continuous research. At the same time, in order to fully study the impact of the 2018 Hokkaido Eastern Iburi earthquake on the Atsuma River channel, it is necessary to incorporate downstream and estuary data.

**Supplementary Materials:** The following are available online at https://www.mdpi.com/article/10.3390/su132313041/s1, Table S1: Average wind speed data at Atsuma station, Table S2: Daily precipitation data, Table S3: Inflow and outflow data at Apporo dam, Table S4: Maximum and minimum temperature data at Atsuma station, Table S5: Observation runoff data at Atsuama Bridge satation, Table S6: Outflow data at Atsuma dam, Table S7: Relative humidity data at Tomakomai station, Table S8: Solar radiation data at Sapporo station, Table S9: Suspended solida at Upstream of construction section, Table S10: The information of dataset.

**Author Contributions:** Methodology, M.N. and Y.C.; software, Y.C.; investigation, M.N. and Y.C.; data curation, M.N. and Y.C.; writing—original draft preparation, Y.C.; writing—review and editing, M.N. and Y.C.; supervision, M.N.; funding acquisition, M.N. All authors have read and agreed to the published version of the manuscript.

**Funding:** This research was funded by the River Center of Hokkaido and the JSPS KAKENHI, grant number JP 20H02404.

**Data Availability Statement:** The data presented in this study are available in supplementary material.

**Acknowledgments:** All authors thank the Hokkaido Regional Development Bureau (MLIT), Hokkaido prefecture, River Center for their help to accomplish this study.

**Conflicts of Interest:** The authors declare no conflict of interest.

## Abbreviations

| Acronym | Description |
| --- | --- |
| SWAT | Soil and Water Assessment Tool |
| LULC | Land Use and Land Cover |
| DEM | Digital Elevation Model |
| USDA | United States Department of Agriculture |
| HRU | Hydrological Response Unit |
| SWAT-CUP | Soil and Water Assessment Tool Calibration and Uncertainty Programs |
| SUFI-2 | Sequential Uncertainty Fitting Procedure Version 2 |
| $R^2$ | Coefficient of Determination |
| NSE | Nash–Sutcliffe Efficiency Coefficient |
| PBIAS | Percent Bias |
| PPMCC | Pearson Product-Moment Correlation Coefficient |

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
