# Peer review of "Analysis of Changes in Land Use/Land Cover and Hydrological Processes Caused by Earthquakes in the Atsuma River Basin in Japan"

_sustainability, doi:10.3390/su132313041_

Round 1
Reviewer 1 Report
Dear authors,
I have gone through your manuscript and I have a number of comments related to the communication of data and methods in particular. I would ask you to look into the details and provide replies and introduce some modifications where needed.
While these are doable issues as you know your data well, I would also ask you to provide comments detailing in how far this manuscript is different from the version you published earlier this year in mdpi/water. I noticed a substantial overlap of wording, and also artwork and data, and what is even more, that other investigation deals with the very same topic at the very same location. However, you neither build on your previous investigation, nor do you cite your previous work, and I wonder why that is the case. I also wonder what happened to the third author of that other study. While he might not be working in your Lab anymore, he seemed to have contributed quite significantly to that original study according to you "author contribution" statement in your other paper published in mdpi/water:
- software, Y.C. and H.O.;
- data curation, M.N., H.O. and Y.C.;
- writing—review and editing, Y.C., M.N. and H.O.;
I would ask you to provide a rebuttal/comments detailing the development of this new manuscript and also provide some insights in how far this current version shows an additional significant and therefore publishable level of novelty. For this very reason, I have to recommend rejection but I would also recommend encouraging re-submission along with provision of more details.
For the remaining manuscript I would ask you to check the following issues:
- The title implies a general approach to measure land-use change and changes in hydrological "processes" (regime?) caused by earthquakes, however, you are investigating a very specific earthquake and the impact in a very specific location, and I do not feel your work shows clearly that this methodology can be applied in different scenarios and act as a sort of methodological blueprint. So, my suggestion would be to either re-phrase the title and make it more specific (and I do realize it then overlaps with the title of the paper published in water), or you add some additional discussion about the wider applicability of your methodological approach. In order to do that, I would expect to see a more abstract workflow diagram and also a discussion about lessons learnt from this particular use case. With that, some additional degree of novelty and originality would be reached which again justifies publication. That is, however, just my personal research view.
- The methodology is quite segmented and nested due to a number of sections and subsections that I feel are not needed considering the small paragraphs. You might want to consider to move things into larger blocks to keep the flow.
- None of your datasets has been adequately described.
-- Please, provide details for your observations data, not just origin, but also dates/acquisition information, formats...
-- What are the image numbers and acquisition dates and other metadata for your L8 scenes?
-- Wow did you conduct processing and data integration?
-- How exactly did you measure classes/areas,
-- What is the nature of the DEM (origin, sensors)
- Why do your map legends say "Illustration"? That makes no sense at all. Why not naming them according to the data that you see?
- You might want to add a box in fig 1 outlining location of fig 3.
- You said that data is available in the supplement. I have not seen that in this submission but I would like to take a look for a proper review.
- The discussion and presentation of results are in principle ok but I miss a discussion about applicability beyond this very location as discussed above. Please provide more insights, I do not feel this suffices.
- Also, please allow an English language editor to have another look at this manuscript as some sentences are quite convoluted and structurally a bit problematic.
Kind Regards.
Author Response
Response to Reviewer 1 Comments
Point 1: While these are doable issues as you know your data well, I would also ask you to provide comments detailing in how far this manuscript is different from the version you published earlier this year in mdpi/water. I noticed a substantial overlap of wording, and also artwork and data, and what is even more, that other investigation deals with the very same topic at the very same location. However, you neither build on your previous investigation, nor do you cite your previous work, and I wonder why that is the case.
I would ask you to provide a rebuttal/comments detailing the development of this new manuscript and also provide some insights in how far this current version shows an additional significant and therefore publishable level of novelty.
Response 1:
Thank you very much for your constructive comments and hard work. First of all, it needs to be emphasized that although the research basin and event of this manuscript are the same as last manuscript, the most core data (land use data), important methods (methods for adjusting parameters) and main conclusions (land use change before and after the earthquake and increase in sediment transport per unit runoff ) are different with last manuscript. Especially with the acquisition and use of land use data after the earthquake and measured sediment transport data, the research process and conclusions of this manuscript are quite different from those of the last manuscript, and the research results of this paper are more meaningful. Which can be said to be a great progress. Next, I will elaborate the differences between this manuscript and last manuscript, as well as the innovations of this manuscript.
The research in the last manuscript was beginned from October 2019. At that time, only the land use data before the earthquake and the measured turbidity data which is highly related to sediment transport were obtained. The land use data after the earthquake and the measured sediment transport data were not obtained. As a result, failed to analyze and quantify the changes in land use before and after the earthquake, and in the process of simulating the sediment transport, it also failed to use the land use data after the earthquake to accurately simulate the sediment transport after the land use change. Only Using the high relation between the sediment transport and the measured turbidity data, manually adjusted the sediment transport parameters after the earthquake to restore the sediment transport after the earthquake as much as possible instead. So as to perform qualitative analysis, after the earthquake, the amount of sediment transport increased significantly. In the abstract and conclusion of last manuscript, it is clearly stated: It is a rough estimate of the sediment transport before and after the earthquake. And in the sediment transport results part, it is also clearly stated: The predictions presented here should be viewed more as qualitative trends, rather than as accurate absolute numerical predictions. As research advances, I got the land use data after the earthquake in March 2021, and the measured sediment transport data was obtained in May 2021. So in the follow-up research, new data was used for new research. First, by comparing the land use data before and after the earthquake, the land changes before and after the earthquake are quantified for the first time. This is an important research result. Secondly, by comparing the measured sediment transport data with the simulation results, the simulation accuracy of the sediment transport process is greatly improved, the rationality of the research is increased, and the simulated sediment transport process is more meaningful. In addition, in the calibration of sediment parameters, because there was no measured sediment transport data during the study of the last manscript, manual parameter calibration method was adopted. However, this manscript use SWAT-CUP software to carry out the sensitivity of sediment parameters analyze and calibration. Making the parameter calibration more accurate and reasonable.
It needs to be emphasized again that the core of this manuscript is the quantification and analysis of land use and land cover changes before and after the earthquake (this work was not carried out in the last manuscript), and then based on the new land use and land cover data to simulate runoff and sediment transport in the basin, and to analysis of changes and reasons for hydrological process changes.
In summary, although the research basin and event of the two manuscript are the same, this manuscript has obvious progress and innovation.
The last manuscript has been cited in the introduction, please see the line: 71.
Point 2: I also wonder what happened to the third author of that other study. While he might not be working in your Lab anymore, he seemed to have contributed quite significantly to that original study according to you "author contribution" statement in your other paper published in mdpi/water:
- software, Y.C. and H.O.;
- data curation, M.N., H.O. and Y.C.;
- writing—review and editing, Y.C., M.N. and H.O.;
Response 2:
Thank you very much for your comments. First, let me briefly introduce the three authors of the last manuscript (Yuechao Chen, Makoto Nakatsugawa and Hiroki Ohashi) to you. I (Yuechao Chen) am a PhD student at Muroran Institute of Technology from October 2019. My supervisor is Professor Makoto Nakatsugaw. Mr. Hiroki Ohashi is an employee of OYO Corporation in Japan who is familiar with the use of the SWAT model and the downloading and sorting of various Japanese data. Mr. Hiroki Ohashi also has conducted research on the Atsumna River basin before. Through the introduction of Professor Makoto Nakatsugawa, I got in touch with Mr. Hiroki Ohashi. As a beginner, during my research process, I meeted many questions about the use of SWAT model and how to download and organize various data. Mr. Hiroki Ohashi answered my questions many times through emails and video conferences. Teaching me many methods and skills, and provided great help to my research. After the first version of the last manuscript was completed, Mr. Hiroki Ohashi reviewed and edited the paper. Therefore, in the last manuscript, Mr. Hiroki Ohashi became the third author. After the completion of the last manuscript, I have been able to independently use the SWAT model and download, organize various data, so in the subsequent research process (download, update, processing of various data and the establishment of new SWAT projects in the basin, simulation process , analysis of results, etc.), I did not get any help from Mr. Hiroki Ohashi anymore (in fact, after the publication of the last paper, I had no contact with Mr. Hiroki Ohashi at all), and after the first draft of this manuscript was completed, I did not ask Mr. Hiroki Ohashi to review and edit. Therefore, Mr. Hiroki Ohashi is no longer a co-author in this manuscript.
Point 3: The title implies a general approach to measure land-use change and changes in hydrological "processes" (regime?) caused by earthquakes, however, you are investigating a very specific earthquake and the impact in a very specific location, and I do not feel your work shows clearly that this methodology can be applied in different scenarios and act as a sort of methodological blueprint. So, my suggestion would be to either re-phrase the title and make it more specific, or you add some additional discussion about the wider applicability of your methodological approach.
Response 3:
Thank you very much for your comments. As you said, this study was focused on the impact of the 2018 Hokkaido Eastern Iburi Earthquake on the land use and hydrological processes in the Atsuma River basin. The research ideas and methods of this study are applicable to different scenarios in similar situations, but the conclusion of this study is only applicable to the impact of 2018 Hokkaido Eastern Iburi Earthquake on the Atsuma River basin. According to your suggestion, the title has been rephrased.
Point 4: In order to do that, I would expect to see a more abstract workflow diagram and also a discussion about lessons learnt from this particular use case. With that, some additional degree of novelty and originality would be reached which again justifies publication.
Response 4:
Thank you very much for your comments. According to your suggestion, the research technology roadmap has been added, and the lessons learned from this particular use case have been added to the conclusion section. Please see figure 5 and line 414.
Point 5: The methodology is quite segmented and nested due to a number of sections and subsections that I feel are not needed considering the small paragraphs. You might want to consider to move things into larger blocks to keep the flow.
Response 5:
Thank you very much for your comments. According to your suggestion, the methodology part has been modified. The updated content was in Line 222.
Point 6: None of your datasets has been adequately described.
-- Please, provide details for your observations data, not just origin, but also dates/acquisition information, formats...
-- What are the image numbers and acquisition dates and other metadata for your L8 scenes?
-- What did you conduct processing and data integration?
-- How exactly did you measure classes/areas,
-- What is the nature of the DEM (origin, sensors)
Response 6:
Thank you very much for your comments. First of all, according to your suggestions, Table 1 has been modified to explain the data used in this study more detail.
Second, the image numbers of the original Landsat 8 data used in this study are:
LC08_L2SP_107030_20150923_20200908_02_T1_QA_PIXEL and LC08_L2SP_107030_20200531_20200820_02_T1_QA_PIXEL。
I downloaded the Landsat 8 data from the United States Geological Survey (USGS) website (https://earthexplorer.usgs.gov/) in March 2021.
The data includes:
LC08_L2SP_107030_20150923_20200908_02_T1_MTL.xml
LC08_L2SP_107030_20150923_20200908_02_T1_QA_PIXEL.TIF
LC08_L2SP_107030_20150923_20200908_02_T1_QA_PIXEL.TIF.enp
LC08_L2SP_107030_20150923_20200908_02_T1_QA_RADSAT.TIF
LC08_L2SP_107030_20150923_20200908_02_T1_QA_RADSAT.TIF.enp
LC08_L2SP_107030_20150923_20200908_02_T1_SR_B1.TIF
LC08_L2SP_107030_20150923_20200908_02_T1_SR_B1.TIF.enp
LC08_L2SP_107030_20150923_20200908_02_T1_SR_B2.TIF
LC08_L2SP_107030_20150923_20200908_02_T1_SR_B2.TIF.enp
LC08_L2SP_107030_20150923_20200908_02_T1_SR_B3.TIF
LC08_L2SP_107030_20150923_20200908_02_T1_SR_B3.TIF.enp
LC08_L2SP_107030_20150923_20200908_02_T1_SR_B4.TIF
LC08_L2SP_107030_20150923_20200908_02_T1_SR_B4.TIF.enp
LC08_L2SP_107030_20150923_20200908_02_T1_SR_B5.TIF
LC08_L2SP_107030_20150923_20200908_02_T1_SR_B5.TIF.enp
LC08_L2SP_107030_20150923_20200908_02_T1_SR_B6.TIF
LC08_L2SP_107030_20150923_20200908_02_T1_SR_B6.TIF.enp
LC08_L2SP_107030_20150923_20200908_02_T1_SR_B7.TIF
LC08_L2SP_107030_20150923_20200908_02_T1_SR_B7.TIF.enp,
LC08_L2SP_107030_20150923_20200908_02_T1_SR_QA_AEROSOL.TIF
LC08_L2SP_107030_20150923_20200908_02_T1_SR_QA_AEROSOL.TIF.enp
LC08_L2SP_107030_20150923_20200908_02_T1_SR_stac.json
LC08_L2SP_107030_20150923_20200908_02_T1_ST_ATRAN.TIF
LC08_L2SP_107030_20150923_20200908_02_T1_ST_ATRAN.TIF.enp
LC08_L2SP_107030_20150923_20200908_02_T1_ST_B10.TIF
LC08_L2SP_107030_20150923_20200908_02_T1_ST_B10.TIF.enp
LC08_L2SP_107030_20150923_20200908_02_T1_ST_CDIST.TIF
LC08_L2SP_107030_20150923_20200908_02_T1_ST_CDIST.TIF.enp
LC08_L2SP_107030_20150923_20200908_02_T1_ST_DRAD.TIF
LC08_L2SP_107030_20150923_20200908_02_T1_ST_DRAD.TIF.enp
LC08_L2SP_107030_20150923_20200908_02_T1_ST_EMIS.TIF
LC08_L2SP_107030_20150923_20200908_02_T1_ST_EMIS.TIF.enp
LC08_L2SP_107030_20150923_20200908_02_T1_ST_EMSD.TIF
LC08_L2SP_107030_20150923_20200908_02_T1_ST_EMSD.TIF.enp
LC08_L2SP_107030_20150923_20200908_02_T1_ST_QA.TIF
LC08_L2SP_107030_20150923_20200908_02_T1_ST_QA.TIF.enp
LC08_L2SP_107030_20150923_20200908_02_T1_ST_stac.json
LC08_L2SP_107030_20150923_20200908_02_T1_ST_TRAD.TIF
LC08_L2SP_107030_20150923_20200908_02_T1_ST_TRAD.TIF.enp
LC08_L2SP_107030_20150923_20200908_02_T1_ST_URAD.TIF
LC08_L2SP_107030_20150923_20200908_02_T1_ST_URAD.TIF.enp
LC08_L2SP_107030_20150923_20200908_02_T1_thumb_large.jpeg
LC08_L2SP_107030_20150923_20200908_02_T1_thumb_large.jpeg.enp
LC08_L2SP_107030_20150923_20200908_02_T1_thumb_small.jpeg
LC08_L2SP_107030_20150923_20200908_02_T1_thumb_small.jpeg.enp
LC08_L2SP_107030_20200531_20200820_02_T1_MTL.xml
LC08_L2SP_107030_20200531_20200820_02_T1_QA_PIXEL.TIF
LC08_L2SP_107030_20200531_20200820_02_T1_QA_PIXEL.TIF.enp
LC08_L2SP_107030_20200531_20200820_02_T1_QA_RADSAT.TIF
LC08_L2SP_107030_20200531_20200820_02_T1_QA_RADSAT.TIF.enp
LC08_L2SP_107030_20200531_20200820_02_T1_SR_B1.TIF
LC08_L2SP_107030_20200531_20200820_02_T1_SR_B1.TIF.enp
LC08_L2SP_107030_20200531_20200820_02_T1_SR_B2.TIF
LC08_L2SP_107030_20200531_20200820_02_T1_SR_B2.TIF.enp
LC08_L2SP_107030_20200531_20200820_02_T1_SR_B3.TIF
LC08_L2SP_107030_20200531_20200820_02_T1_SR_B3.TIF.enp
LC08_L2SP_107030_20200531_20200820_02_T1_SR_B4.TIF
LC08_L2SP_107030_20200531_20200820_02_T1_SR_B4.TIF.enp
LC08_L2SP_107030_20200531_20200820_02_T1_SR_B5.TIF
LC08_L2SP_107030_20200531_20200820_02_T1_SR_B5.TIF.enp
LC08_L2SP_107030_20200531_20200820_02_T1_SR_B6.TIF
LC08_L2SP_107030_20200531_20200820_02_T1_SR_B6.TIF.enp
LC08_L2SP_107030_20200531_20200820_02_T1_SR_B7.TIF
LC08_L2SP_107030_20200531_20200820_02_T1_SR_B7.TIF.enp,
LC08_L2SP_107030_20200531_20200820_02_T1_SR_QA_AEROSOL.TIF
LC08_L2SP_107030_20200531_20200820_02_T1_SR_QA_AEROSOL.TIF.enp
LC08_L2SP_107030_20200531_20200820_02_T1_SR_stac.json
LC08_L2SP_107030_20200531_20200820_02_T1_ST_ATRAN.TIF
LC08_L2SP_107030_20200531_20200820_02_T1_ST_ATRAN.TIF.enp
LC08_L2SP_107030_20200531_20200820_02_T1_ST_B10.TIF
LC08_L2SP_107030_20200531_20200820_02_T1_ST_B10.TIF.enp
LC08_L2SP_107030_20200531_20200820_02_T1_ST_CDIST.TIF
LC08_L2SP_107030_20200531_20200820_02_T1_ST_CDIST.TIF.enp
LC08_L2SP_107030_20200531_20200820_02_T1_ST_DRAD.TIF
LC08_L2SP_107030_20200531_20200820_02_T1_ST_DRAD.TIF.enp
LC08_L2SP_107030_20200531_20200820_02_T1_ST_EMIS.TIF
LC08_L2SP_107030_20200531_20200820_02_T1_ST_EMIS.TIF.enp
LC08_L2SP_107030_20200531_20200820_02_T1_ST_EMSD.TIF
LC08_L2SP_107030_20200531_20200820_02_T1_ST_EMSD.TIF.enp
LC08_L2SP_107030_20200531_20200820_02_T1_ST_QA.TIF
LC08_L2SP_107030_20200531_20200820_02_T1_ST_QA.TIF.enp
LC08_L2SP_107030_20200531_20200820_02_T1_ST_stac.json
LC08_L2SP_107030_20200531_20200820_02_T1_ST_TRAD.TIF
LC08_L2SP_107030_20200531_20200820_02_T1_ST_TRAD.TIF.enp
LC08_L2SP_107030_20200531_20200820_02_T1_ST_URAD.TIF
LC08_L2SP_107030_20200531_20200820_02_T1_ST_URAD.TIF.enp
LC08_L2SP_107030_20200531_20200820_02_T1_thumb_large.jpeg
LC08_L2SP_107030_20200531_20200820_02_T1_thumb_large.jpeg.enp
LC08_L2SP_107030_20200531_20200820_02_T1_thumb_small.jpeg
LC08_L2SP_107030_20200531_20200820_02_T1_thumb_small.jpeg.enp
Secondly, regarding the processing of land use data, this study used ENVI software to process the original data of Landset 8 through stacke, band fusion and clip, classification, rasterized, projection, etc., and obtained the land use data before and after the earthquake which can be used by the SWAT model.
Then, open the processed land use data through GIS software, and then through the statistical function of GIS software, the area of different land types in the basin can be obtained.
Finally, the nature of DEM data is satellite elevation data. In the process of using the SWAT model to build a watershed hydrological model, DEM data needs to be input.
Point 7: Why do your map legends say "Illustration"? That makes no sense at all. Why not naming them according to the data that you see?
You might want to add a box in fig 1 outlining location of fig 3.
Response 7:
Thank you very much for your comments. According to your suggestion, Figure 1 has been modified, and the area of Figure 3 is also marked in Figure 1. Please check the new Figure 1.
Point 8: You said that data is available in the supplement. I have not seen that in this submission but I would like to take a look for a proper review.
Response 8:
Thank you very much for your comments. The measured data used in this research has been submitted as supplementary materials when the paper was submitted for the first time. Please do not worry.
Point 9: The discussion and presentation of results are in principle ok but I miss a discussion about applicability beyond this very location as discussed above. Please provide more insights, I do not feel this suffices.
Response 9:
Thank you very much for your comments. This study focused on the land use change in the Atsuma River basin after 2018 Hokkaido Eastern Iburi Earthquake, and its impact on the hydrological process of the basin. The research ideas and methods of this study are applicable to different scenarios in similar situations, but the conclusions of this study are only applicable to the impact of 2018 Hokkaido Eastern Iburi Earthquake on the Atsuma River basin. According to your suggestion, the conclusion part has been modifitied.
Point 10: Also, please allow an English language editor to have another look at this manuscript as some sentences are quite convoluted and structurally a bit problematic.
Response 10:
Thank you very much for your comments. Regarding the English wording and grammar of this manuscript, we carefully revised it verbatim; secondly, with the help of professional English editors, we carefully, conscientiously and deeply modified the full text again. We believe that the quality of English writing has been improved greatly.
Finally, I would like to express my heartfelt thanks again to you for your wonderful comments and hard work.

Reviewer 2 Report
This paper gives a evaluation on the changes in land use change after the Hokkaido earthquake and its impact on the runoff and sediment transport process using remote sensing interpretation and hydrological calculation. The structure is complete and the logic is resonable. However, the innovation of the paper is not clear yet. I would give a recommendation of Major Revision.
For section 2.2.1, the subtitle is about the earthquake, so more info about the earthquake or its impact is expected to spread here. But most contents of this section is about topographical interpretation results.
Some figures, especially after Figure. 5, can be more exquisite.
Again, the innovation of this paper is not clear yet. More description about the breakthough the authors made is suggested to be added here.
Author Response
Response to Reviewer 2 Comments
Point 1: This paper gives a evaluation on the changes in land use change after the Hokkaido earthquake and its impact on the runoff and sediment transport process using remote sensing interpretation and hydrological calculation. The structure is complete and the logic is resonable. However, the innovation of the paper is not clear yet. I would give a recommendation of Major Revision.
Response 1:
First of all, thank you very much for your wonderful comments and hard work. Your comments has played a vital role in improving the quality of this paper. I would like to express my heartfelt thanks again to you。
Point 2: For section 2.2.1, the subtitle is about the earthquake, so more info about the earthquake or its impact is expected to spread here. But most contents of this section is about topographical interpretation results.
Response 2:
Thank you very much for your comments. The content about earthquake and its impact have been added in section 2.2.1. The added content is: In terms of human and property damage, the earthquake killed 41 people in Hokkaido and injured at least 692, including 13 serious injuries and 679 minor injuries. The earthquake also damaged at least 2,508 buildings, several roads and buried several cars by mudslides. In terms of production and daily life, the earthquake caused damage to equipment of Tomato Atsuma Electric Power Station, the largest thermal Power plant in Hokkaido, and Unit 2 of Onbetsu Electric Power Station in Ibetsu, Kushiro. Hokkaido lost more than half of its power supply, resulting in a total of 2.95 million households across Hokkaido without electricity. Running water and telecommunications were also suspended in some parts of Hokkaido, supermarkets and convenience stores were in short supply, and Hokkaido's 1,800 schools were temporarily closed. Hokkaido's agriculture, forestry, aquaculture and aquaculture were serious economic losses by the power outages and geological disaster cased by earthquake . Several Hokkaido plants, including SUMCO Chitosa fabs, CALBEE and Sapporo Breweries, have suspended production.
Point 3: Some figures, especially after Figure. 5, can be more exquisite.
Response 3:
Thank you very much for your comments. According to your suggestion, all the pictures in this paper have been modified and improved, the new pictures can display information more clearly and accurately.
Point 4: Again, the innovation of this paper is not clear yet. More description about the breakthough the authors made is suggested to be added here.
Response 4:
Thank you very much for your comments. About the innovation of this study. First of all, there are many studies on the 2018 Hokkaido Eastern Iburi Earthquake and other earthquakes in history, but there are few studies on the land use and land cover change caused by the earthquake and the changes in runoff and sediment transport processes in the basin after the earthquake. Therefore, this study has significance for understanding the impact of earthquakes on human and the environment. Secondly, most of the previous studies on land use and land cover change focused on land use and land cover change caused by rapid urbanization, deforestation and agriculture. There were few studies on land use and land cover caused by natural disasters including earthquakes. This study is of significance for a more comprehensive understanding of land use and land cover change. Third, the land use and land cover change caused by earthquakes and its landslides are different with other ways, the impact of earthquakes and its landslides are not only fast, dramatic but also producing considerable sediment, debris, trees, etc., which are often close to the river and unstable. This increases the difficulty of studying runoff and sediment transport processes, and makes this study more meaningful. In summary, this research has obvious innovation and significance.

Reviewer 3 Report
The manuscript is well-conducted and, without doubts it covers an interesting topic. The topic and the research conducted is of high degree of novelty and the material&methods is correct.
The results&discussion and the conclusions are related with the hypothesis of the manuscript.
This manuscript contributes with the scientific knowledge.
Author Response
Response to Reviewer 3 Comments
Thank you very much for your comments and hard work.
Regarding the English spell checking of this manuscript, based on your valuable revision comments, we carefully revised it verbatim; secondly, with the help of professional English editors, we carefully, conscientiously and deeply modified the full text again. We believe that the quality of English writing has been improved greatly.
Thank you so much again.

Reviewer 4 Report
Reviewer’s Report on the manuscript entitled:
Analysis of abrupt land use change and changes in the hydrological process in river basins caused by earthquakes
The authors investigated the land cover land use change before and after the 2018 Hokkaido Eastern Iburi earthquake and its impact on runoff and sediment transport in Atsuma River Basin in Japan using the Soil and Water Assessment Tool (SWAT) model. Although the topic and results are interesting, the presentation and grammar need significant improvement. Below, I have listed my comments.
Firstly, I suggest the use of Land Use Land Cover (LULC) across the manuscript because earthquake has changed some forested areas to bare land and that is land cover change. On the other hand, human activity, such as urbanization results in land use change. So, when you use the widely use LULC, it will make more sense. This also applies in the title too. So, in the title say land use land cover change. I would suggest changing the title of the manuscript to something like:
Analysis of Land Use Land Cover and Hydrological Changes Caused by Earthquakes in Atsuma River Basin in Japan
Lines 11,12. Grammar issue. Please rewrite the sentence.
Line 16. Is should be “is” not “are”.
Lines 17, 358. 4.42 times what? It should be “4.42 times more than before”.
Line 18. “…; whereas, …” Insert semicolon and comma.
Line 20, 21. Grammar issue. Please rewrite.
Line 26. 6 what? Richter?
Line 27. “and December” not “to December”.
Lines 28, 29. Rewrite the sentence.
Line 34. “earthquakes”.
Line 46. “cannot” is one word.
Line 99. Please also mention
“Robust techniques other than SWAT models are also applied to investigate the relationships between climate change and streamflow. For example, Ghaderpour et al. [https://doi.org/10.1016/j.ejrh.2021.100847] applied the least-squares cross-wavelet analysis to show the impact of climate change on snowmelt and streamflow in Athabasca River Basin in Canada. Zerouali et al. [https://doi.org/10.3390/w13212946] also applied the cross-wavelet transform to assess the response of daily rainfall and karst spring discharge for the Sebaou River basin in northern Algeria.”
Figures. Please ensure all figures have a resolution of at least 300dpi.
Figures 1,2,3. Please enlarge the font size of the texts.
Line 101. It should be “Landsat” not “land set”.
Line 108. Please insert the minute symbol for the coordinates.
Line 138. Please add the reference number instead of the year for Zhou et al.
Line 171. It should be “for 2009-2020 from…”
Line 173. It should be “for 2015-2020 from…”
Line 178. This is the first time you use the term LULC! As per my earlier comment, this term has to be defined and used from the beginning of the manuscript, including the title.
Line 185. Replace “of a” with “at”.
Equation (1). Should not it be SWi instead of SWo?
Please insert the y-axis label in Figure 12. Please remove the square border in this figure and other figures.
Line 378. “… significantly from…”
Line 392. “was” not “were”
The limitations of the study must be mentioned in the Conclusion section.
Please also add an acronym table at the end of the manuscript listing all acronyms used.
Thank you for your contribution
Regards,
Author Response
Response to Reviewer 4 Comments
Point 1: Firstly, I suggest the use of Land Use Land Cover (LULC) across the manuscript because earthquake has changed some forested areas to bare land and that is land cover change. On the other hand, human activity, such as urbanization results in land use change. So, when you use the widely use LULC, it will make more sense. This also applies in the title too. So, in the title say land use land cover change. I would suggest changing the title of the manuscript to something like:
Analysis of Land Use Land Cover and Hydrological Changes Caused by Earthquakes in Atsuma River Basin in Japan
Response 1:
First of all, thank you very much for your constructive comments and hard work. Your comments has played a vital role in improving the quality of this paper. I would like to express my heartfelt thanks again to you.
According to your suggestion, “Land use” has been all replaced by “Land use and land cover” across this manuscript, and the title has been changed to: Analysis of Land Use and Land Cover and Hydrological Changes Caused by Earthquakes in Atsuma River Basin in Japan. The use of land use and land cover and new title really makes this article more reasonable and meaningful.
Point 2: Lines 11,12. Grammar issue. Please rewrite the sentence.
Response 2:
Thank you very much for your comments. According to your suggestion, it has been modified. Please see line: 10, 11, 12, 13.
Point 3: Line 16. Is should be “is” not “are”.
Response 3:
Thank you very much for your comments. According to your suggestion, it has been modified. Please see line: 16.
Point 4: Lines 17, 358. 4.42 times what? It should be “4.42 times more than before”.
Response 4:
Thank you very much for your comments. According to your suggestion, it has been modified. Please see line: 18.
Point 5: Line 18. “…; whereas, …” Insert semicolon and comma.
Response 5:
Thank you very much for your comments. According to your suggestion, it has been modified. Please see line: 19.
Point 6: Line 20, 21. Grammar issue. Please rewrite.
Response 6:
Thank you very much for your comments. According to your suggestion, it has been modified. Please see line: 20, 21.
Point 7: Line 26. 6 what? Richter?
Response 7:
Thank you very much for your comments. The Moment magnitude scale is used to measure earthquakes here, it has been modified. Please see line: 26.
Point 8: Line 27. “and December” not “to December”.
Response 8:
Thank you very much for your comments. According to your suggestion, it has been modified. Please see line: 27.
Point 9: Line 34. “earthquakes”.
Response 9:
Thank you very much for your comments. According to your suggestion, it has been modified. Please see line: 31.
Point 10: Line 46. “cannot” is one word.
Response 10:
Thank you very much for your comments. According to your suggestion, it has been modified. Please see line: 43.
Point 11: Line 99. Please also mention
“Robust techniques other than SWAT models are also applied to investigate the relationships between climate change and streamflow. For example, Ghaderpour et al. [https://doi.org/10.1016/j.ejrh.2021.100847] applied the least-squares cross-wavelet analysis to show the impact of climate change on snowmelt and streamflow in Athabasca River Basin in Canada. Zerouali et al. [https://doi.org/10.3390/w13212946] also applied the cross-wavelet transform to assess the response of daily rainfall and karst spring discharge for the Sebaou River basin in northern Algeria.”
Response 11:
Thank you very much for your comments. I have carefully read the two papers you mentioned, and added the response content in this manuscript. Please see line: 94.
Point 12: Figures. Please ensure all figures have a resolution of at least 300dpi.
Response 12:
Thank you very much for your comments. The pictures of full manuscript have been modified with the requirements (minimum 1000 pixels width/height, or a resolution of 300 dpi or higher) of the journal.
Point 13: Figures 1,2,3. Please enlarge the font size of the texts.
Response 13:
Thank you very much for your comments. According to your suggestion, it has been modified. Please see new figure1, 2 3.
Point 14: Line 101. It should be “Landsat” not “land set”.
Response 14:
Thank you very much for your comments. According to your suggestion, it has been modified. Please see line: 101.
Point 15: .Line 108. Please insert the minute symbol for the coordinates.
Response 15:
Thank you very much for your comments. According to your suggestion, it has been modified. Please see line: 108.
Point 16: Line 138. Please add the reference number instead of the year for Zhou et al.
Response 16:
Thank you very much for your comments. According to your suggestion, it has been modified. Please see line: 141, 148.
Point 17: Line 171. It should be “for 2009-2020 from…”
Line 173. It should be “for 2015-2020 from…”
Response 17:
Thank you very much for your comments. According to your suggestion, it has been modified. Please see line: 203, 205.
Point 18: Line 178. This is the first time you use the term LULC! As per my earlier comment, this term has to be defined and used from the beginning of the manuscript, including the title.
Response 18:
Thank you very much for your comments. According to your suggestion, the full manuscript has been modified.
Point 19: Equation (1). Should not it be SWi instead of SWo?
Response 19:
Thank you very much for your comments. After checking again, the SWo in the formula is correct. There are mistakes with my previous interpretation of the formula, which brings you questions. It has been modified, please see line: 176.
Point 20: Please insert the y-axis label in Figure 12. Please remove the square border in this figure and other figures.
Response 20:
Thank you very much for your comments. According to your suggestion, it has been modified. Please see Figure 13.
Point 21: Line 378. “… significantly from…”
Response 21:
Thank you very much for your comments. According to your suggestion, it has been modified. Please see line: 372.
Point 22: Line 392. “was” not “were”
Response 22:
Thank you very much for your comments. According to your suggestion, it has been modified. Please see line: 386.
Point 23: The limitations of the study must be mentioned in the Conclusion section.
Response 23:
Thank you very much for your comments. The limitations of this study have been explained in the conclusion section. Please see line: 404.
Point 24: Please also add an acronym table at the end of the manuscript listing all acronyms used.
Response 24:
Thank you very much for your comments. A acronym table has been added at the end of the text, please see table 8.
Finally, I would like to express my heartfelt thanks again to you for your wonderful comments and hard work.

Round 2
Reviewer 1 Report
Dear authors,
thank you for addressing my comments and for providing detailed insights into the development of this manuscript, which I feel is needed -- not just for the review process, but also in order to emphasize novelty and original research later on.
I have certainly no objections as this development has been clarified in much detail, and beyond that it lies at the discretion of the publisher. Again, I appreciate your detailed response.
Regarding data presentation, however, I still have minor issues.
First of all, I do not think you have modified your legends, as they still say 'illustration' in most of your figures where you show maps. So, please change that to something common in cartographic presentation and say "slopes, degree" , or "land use classes" and so on, instead of "illustration", for all your figures.
Secondly, while I appreciate that you changed your title according to the actual contents, it reads a bit 'bumpy'. You might want to consider something like this (totally up to you of course):
Analysis of Changes in Land Use/Land Cover and Hydrological Regime Caused by Earthquakes in the Atsuma River Basin in Japan.
Thirdly, please provide the list of used data to the reader, not only to the reviewer. I would strongly suggest to make a table (either inline or as supplement) detailing:
Sensor platform | Image number path/row | acquisition date and time |
Metadata files (txt, json) and previews (jpegs) are not part of that list. Neither are formats (tif, enp, ...). Just the image numbers along with their imaging dates and location as well as spectral (or other relevant) information.
Thank you and kind regards.
Author Response
Response to Reviewer 1 Comments
Point 1: thank you for addressing my comments and for providing detailed insights into the development of this manuscript, which I feel is needed -- not just for the review process, but also in order to emphasize novelty and original research later on.
I have certainly no objections as this development has been clarified in much detail, and beyond that it lies at the discretion of the publisher. Again, I appreciate your detailed response.
Response 1:
Thank you again for your wonderful comments and hard work. Your comments has played a very important role in improving the quality of this paper.
Point 2: Regarding data presentation, however, I still have minor issues.
First of all, I do not think you have modified your legends, as they still say 'illustration' in most of your figures where you show maps. So, please change that to something common in cartographic presentation and say "slopes, degree" , or "land use classes" and so on, instead of "illustration", for all your figures.
Response 2:
Thank you very much for your comments. According to your suggestion, all the pictures which including 'illustration' in this paper have been modified. Please see new figure 2, 4, 6.
Point 3: Secondly, while I appreciate that you changed your title according to the actual contents, it reads a bit 'bumpy'. You might want to consider something like this (totally up to you of course):
Analysis of Changes in Land Use/Land Cover and Hydrological Regime Caused by Earthquakes in the Atsuma River Basin in Japan.
Response 3:
Thank you very much for your comments. According to your suggestion, the title has been changed to: Analysis of Changes in Land Use/Land Cover and Hydrological Regime Caused by Earthquakes in the Atsuma River Basin in Japan. The new title really more reasonable. Thank you again.
Point 4: Thirdly, please provide the list of used data to the reader, not only to the reviewer. I would strongly suggest to make a table (either inline or as supplement) detailing: Sensor platform | Image number path/row | acquisition date and time |
Metadata files (txt, json) and previews (jpegs) are not part of that list. Neither are formats (tif, enp, ...). Just the image numbers along with their imaging dates and location as well as spectral (or other relevant) information.
Response 4:
Thank you very much for your comments. According to your suggestion. First, I have modified the table 1 in section 2.4 to add information about the dataset. Second, I added a table called "The information of dataset" in the supplementary material , which detailing: Source and download path, Sensor platform, Data number, Imaging date, Spectral, Location and other information of data.
Finally, I would like to express my heartfelt thanks again to you for your wonderful comments and hard work.

Reviewer 2 Report
I read the revised version and the authors comments. Most of my concerns have been addressed and the paper can be accepted for publication.
Author Response
Response to Reviewer 2 Comments
Point 1: I read the revised version and the authors comments. Most of my concerns have been addressed and the paper can be accepted for publication.
Response 1:
Thank you again for your constructive comments and hard work. Your comments has played a very important role in improving the quality of this paper.

Reviewer 4 Report
I would like to thank the authors for addressing my comments. The manuscript looks much better now, but please carefully proofread the article if accepted by the editor. I have found some more minor editorial comments:
Line 99. Should be "Landsat" not "landset".
Line 157. "..., there are large..."
Line 180. "Figures 7-10 show..."
Thank you for your contribution
Regards,
Author Response
Response to Reviewer 4 Comments
Point 1: Line 99. Should be "Landsat" not "landset".
Response 1:
First of all, thank you again for your constructive comments and hard work. Your comments has played a vital role in improving the quality of this paper. I would like to express my heartfelt thanks again to you.
According to your suggestion, it has been modified. Please see line: 101.
Point 2: Line 157. "..., there are large..."
Response 2:
Thank you very much for your comments. According to your suggestion, it has been modified. Please see line: 163.
Point 3: Line 180. "Figures 7-10 show..."
Response 3:
Thank you very much for your comments. According to your suggestion, it has been modified. Please see line: 287.
Finally, I would like to express my heartfelt thanks again to you for your wonderful comments and hard work.
